# *CARD8* and *IL1B* Polymorphisms Influence MRI Brain Patterns in Newborns with Hypoxic-Ischemic Encephalopathy Treated with Hypothermia

**DOI:** 10.3390/antiox10010096

**Published:** 2021-01-12

**Authors:** Katarina Esih, Katja Goričar, Zvonka Rener-Primec, Vita Dolžan, Aneta Soltirovska-Šalamon

**Affiliations:** 1Division of Paediatrics, Department of Child, Adolescent and Developmental Neurology, University Medical Centre Ljubljana, 1000 Ljubljana, Slovenia; katarina.esih@gmail.com (K.E.); zvonka.renerprimec@kclj.si (Z.R.-P.); 2Pharmacogenetics Laboratory, Institute of Biochemistry and Molecular Genetics, Faculty of Medicine, University of Ljubljana, 1000 Ljubljana, Slovenia; katja.goricar@mf.uni-lj.si; 3Faculty of Medicine, University of Ljubljana, 1000 Ljubljana, Slovenia; 4Division of Pediatrics, Department of Neonatology, University Medical Centre Ljubljana, 1000 Ljubljana, Slovenia

**Keywords:** inflammation, oxidative stress, polymorphism, hypoxic-ischemic encephalopathy, MRI, newborn

## Abstract

Inflammation and oxidative stress are recognized as important contributors of brain injury in newborns due to a perinatal hypoxic-ischemic (HI) insult. Genetic variability in these pathways could influence the response to HI and the outcome of brain injury. The aim of our study was to evaluate the impact of common single-nucleotide polymorphisms in the genes involved in inflammation and response to oxidative stress on brain injury in newborns after perinatal HI insult based on the severity and pattern of magnetic resonance imaging (MRI) findings. The DNA of 44 subjects was isolated from buccal swabs. Genotyping was performed for *NLRP3* rs35829419, *CARD8* rs2043211, *IL1B* rs16944, *IL1B* rs1143623, *IL1B* rs1071676, *TNF* rs1800629, *CAT* rs1001179, *SOD2* rs4880, and *GPX1* rs1050450. Polymorphism in *CARD8* was found to be protective against HI brain injury detected by MRI overall findings. Polymorphisms in *IL1B* were associated with posterior limb of internal capsule, basal ganglia, and white matter brain patterns determined by MRI. Our results suggest a possible association between genetic variability in inflammation- and antioxidant-related pathways and the severity of brain injury after HI insult in newborns.

## 1. Introduction

Perinatal hypoxic-ischemic encephalopathy (HIE) results from compromised blood flow and/or oxygen delivery to the newborn baby’s brain and is one of the leading causes of long-term neurological disability in children [1]. The incidence of HIE in the developed countries remains around 1–3 per 1000 term newborns over the last decade [1,2]. The most common causes of HIE include placental abruption, cord prolapse, uterine rupture, and breech presentation [3]. The only established therapy for HIE, therapeutic hypothermia (HT), implemented in 2010, improves survival and lowers long-term disability rates, but it is only partially effective [4]. Two methods of cooling have been used in clinical trials—selective head cooling and whole-body cooling. Whole-body cooling is recommended preferentially because it is easier to set up and use, less expensive, provides better access to EEGs, and is more readily available. Although studies suggest that hypothermia exerts a neuroprotective effect by altering several different molecular pathways and affects metabolism, brain perfusion, the release of excitatory amino acids, apoptosis, and antioxidant and inflammatory processes [5,6,7,8], it is not known why some infants benefit from HT while others do not.

The pathophysiological processes after hypoxic-ischemic (HI) insult occur in two main phases: the primary energy failure, followed by the latent period when a brief restoration of blood flow occurs [1,3], and the secondary energy failure [3] during which inflammation and overproduction of reactive oxygen species (ROS) are important for brain injury [9,10]. Tissue destruction contributes further to inflammation through activation of inflammasome, an intracellular multiprotein complex. Its main components are NLRP3, CARD8, and caspase-1. The activated inflammasome promotes maturation and secretion of inflammatory cytokines, mainly interleukin 1β (IL-1β) and tumor necrosis factor (TNFα) [11], which represent early response cytokines secreted by cells in the central nervous system (CNS) [12]. Parallel to inflammation, an overproduction of ROS leads to oxidative stress that is counteracted by antioxidant enzymes. The most important among them are manganese superoxide-dismutase (MnSOD), glutathione peroxidase (GPX), and catalase (CAT) [13]. MnSOD converts superoxide anion into molecular oxygen and hydrogen peroxide [14], whereas GPX and CAT reduce hydrogen peroxide, preventing the formation of hydroxyl radicals [15,16].

Polymorphisms in inflammatory and antioxidant genes can affect protein activity or gene expression by influencing transcription factor or miRNA binding [17]. Genetic factors could therefore alter the subsequent processes of tissue inflammation and destruction following HIE. Consequently, this could also contribute to the outcome after HIE and lead to different patterns and severity of brain injury [18,19] and could be one of the possible mechanisms for the lower effectiveness of the hypothermia treatment in some patients.

Magnetic resonance imaging (MRI) is used early after the acute HI insult to assess the pattern and severity of brain injury and provides objective measurement of early brain damage that can be at least partially predictive for later neurological outcome [20]. In order to predict better the clinical outcome after HIE, in addition to MRI, serum, cerebrospinal fluid, and other biomarkers have been investigated [21,22]. Neuron-specific enolase serum concentration or phosphorylated axonal neurofilament heavy chain protein concentration was associated with MRI changes after HIE [21,23]. IL-1β and Il-6 concentration was also proposed as a biomarker of long-term outcome after HIE [21,22]. However, the role of genetic variability in the outcome after HIE is not well established. Most studies focused on long-term outcome, especially on susceptibility for cerebral palsy [18,24,25,26,27]. *MTHFR* polymorphisms were also proposed as biomarkers of brain injury determined using MRI or other imaging methods [28,29,30]. Among genes associated with inflammation and oxidative stress, only *NOS3* polymorphisms were associated with brain injury on MRI after HIE [31,32], but in these studies, the MRI classification was performed only at two years of age. The association between genetic variability of inflammatory and antioxidant genes and early MRI findings has not been addressed yet.

The aim of our study was to evaluate the association between the severity and a pattern of MRI findings and polymorphisms in genes of antioxidant and inflammatory pathways involved in the pathogenesis after HIE.

## 2. Subjects and Methods

### 2.1. Study Population

Newborns with moderate and severe HIE were retrospectively identified from the electronic database of the neonatal intensive care unit at the University Children’s Hospital Ljubljana. Inclusion criteria were as follows: infants born between 2007 and 2019, with ≥36 weeks of gestation and who underwent treatment with therapeutic hypothermia due to perinatal asphyxia (5 min Apgar score ≤ 5, pH ≤ 7.0, base deficit ≥16 mmol/L, or resuscitation 10 min after birth) [33]. Evidence of moderate or severe encephalopathy was distinguished by the Sarnat and Sarnat clinical scoring system [34]. Whole-body cooling was started within 6 h after birth and continued for 72 h. After 72 h, the babies were rewarmed gradually to 36.5 °C. Brain MRI was performed in all within the first week after birth (between fourth and seventh day) as a part of routine clinical care (therapeutic hypothermia protocol).

Among 84 children who were treated with total body hypothermia at our center, 40 were excluded due to the following reasons: newborns who did not undergo MRI imaging, subjects where informed consent could not be obtained due to death, rejection by parents or legal guardians to participate in the study, or insufficient contact data availability.

### 2.2. MR Imaging Protocol

MRI scans were performed on Siemens 1.5-T Avanto or 3.0-T Trio (Siemens Medical, Erlangen, Germany) scanners. The standard MRI protocol included axial T1-weighted images or inversion recovery-weighted images, T2-weighted images, and diffusion-weighted images. The custom diffusion sequence consisted of 2 × 2 × 2 mm voxels; 9300 ms repetition time; 96 ms echo time; 1710 Hz/Px; and 2 b-values, 0 and 1000. Two reviewers blinded to patient outcomes reviewed all MRIs and classified all images using the score described below. The interobserver variability was estimated, and in case of disagreement, consensus was obtained with a third blinded reviewer.

### 2.3. MR Scoring System

The patterns of brain injury were classified according to the Rutherford classification [35]. The pattern and severity of injury in the following regions were evaluated and scored: posterior limb of internal capsule (PLIC), thalamus and basal ganglia, white matter, and cortex as previously described in the literature [35,36,37]. Additionally, pattern and severity of brainstem injury were analyzed—lesions at mesencephalon and in pons. An overall assessment was determined by adding up all five regional subscores, and classified as normal, mild, moderate, or severe injury [38]. Subjects were classified as normal if there was no injury seen on MRI. If there were mild, moderate, or severe MRI findings in one, two, or all of the assessed regions, overall assessment was defined as mild, moderate, or severe, respectively. We also transformed the scores into binary variables and compared subjects with normal/mild to subjects with moderate/severe pathological MRI findings.

### 2.4. DNA Extraction and Genotyping

Genomic DNA was extracted from buccal swab samples using QIAamp DNA Mini Kit (Qiagen) according to the manufacturer’s instructions. Common single-nucleotide polymorphisms (SNPs) with experimentally or in silico predicted function in antioxidant (*SOD2* rs4880, *CAT* rs1001179, *GPX1* rs1050450) and inflammatory (*NLRP3* rs35829419, *CARD8* rs2043211, *IL1B* rs1143623, *IL1B* rs16944, *IL1B* rs1071676, *TNF* rs1800629) pathways were genotyped using fluorescent-based competitive allele-specific polymerase chain reaction (KASP, LGC Genomics, UK) according to the manufacturer’s instructions.

The study was approved by the Republic of Slovenia National Medical Ethics Committee and informed consent was obtained from all the participants’ parents or legal guardians before inclusion in the study.

### 2.5. Statistical Analysis

Median with interquartile range (25–75%) and frequencies were used to describe continuous and categorical variables, respectively. Dominant genetic model was used in all analyses. Deviation from Hardy–Weinberg equilibrium (HWE) was assessed using the standard Χ^2^ test. Logistic regression was used to evaluate the association of common SNPs with MRI brain patterns, and odds ratios (ORs) with corresponding 95% confidence intervals (CIs) were calculated. Fisher’s exact test was used for comparison of dependent variables with more than two categories. As nine SNPs were investigated, Bonferroni correction was used to account for multiple comparisons: *p*-values below 0.006 were considered statistically significant, while *p*-values between 0.006 and 0.050 were considered nominally significant. All statistical tests were two-sided. For a polymorphism with minor allele frequency of 0.30, this study had 80% power to detect ORs of 5.9 or more. Power calculation was conducted by the PS Power and sample size calculations, version 3.0 [39]. The statistical analysis was performed using IBM SPSS Statistics version 21.0 (IBM Corporation, Armonk, NY, USA).

## 3. Results

In total, 44 newborns met all the inclusion criteria. Clinical characteristics of the participants are presented in Table 1. The median gestational age was 40 (38–40) gestational weeks. The study group consisted of 35 (56.8%) boys and 19 (43.2%) girls (Table 1).

MRI overall assessment showed moderate to severe level of injury in 18 patients (41%) and normal to mild MRI findings in 26 patients (59%) (Table 2). MRI analysis showed that the rates of moderate to severe damage varied among different brain regions (Table 2). The highest number of injuries was documented in PLIC (N = 20, 45.4%). Basal ganglia were moderately to severely damaged in 16 patients (36.4%) and white matter in 18 patients (41%). The cortex and the brainstem had the lowest frequency of moderate–severe damage on MRI (10 and 9 patients, respectively) (Table 2). None of the clinical criteria were associated either with the severity of injury on the overall MRI findings (all *p* > 0.05) or with brain injury patterns (all *p* > 0.05).

### 3.1. Association of Common Polymorphisms with Overall MRI Findings

The genotype frequencies of common polymorphisms in the antioxidant and inflammatory pathways are shown in Appendix A. All of the investigated polymorphisms were in Hardy–Weinberg equilibrium (Appendix A). For *IL1B* rs16944, the polymorphic C allele is more frequent in European populations and was chosen as the reference allele in further analyses.

#### 3.1.1. Antioxidant Pathway

When comparing all MRI pattern groups, carriers of at least one polymorphic *CAT* rs1001179 T allele were more likely than noncarriers to have a normal MRI according to overall assessment (*p* = 0.034). When we grouped patients with normal–mild and moderate–severe injury together, none of the investigated polymorphisms in the common antioxidant genes showed a statistically significant or nominally significant association with the severity of brain damage on MRI (Table 3).

#### 3.1.2. Inflammatory Pathway

Carriers of at least one polymorphic *CARD8* rs2043211 T allele were less likely than noncarriers to have severe MRI damage *(p* = 0.048) and moderate–severe damage (OR = 0.26, 95% CI = 0.07–0.94, *p* = 0.040). Additionally, the polymorphic *IL1B* rs1143623 C allele showed nominally significant association with increased moderate–severe damage on MRI (OR = 3.78, 95% CI = 1.06–13.45, *p* = 0.040).

### 3.2. MRI Brain Patterns

*IL1B* rs1143623 showed a nominally significant association with higher damage of basal ganglia according to MRI (OR = 3.96, 95% CI = 1.07–14.67, *p* = 0.039).

Two polymorphisms in *IL1B* were associated with higher PLIC damage: *IL1B* rs1143623 (OR = 5.67, 95% CI = 1.54–20.79, *p* = 0.009) and *IL1B* rs16944 (OR = 3.71, 95% CI = 1.06–12.97, *p* = 0.040). On the other hand, *IL1B* rs1071676 was associated with lower white matter damage (OR = 0.18, 95% CI = 0.05–0.68, *p* = 0.011) (Table 4). Additionally, *CARD8* rs2043211 was associated with lower brainstem and cortex damage (OR = 0.07, 95% CI = 0.01–0.66, *p* = 0.020 and OR = 0.15, 95% CI = 0.03–0.84, *p* = 0.031, respectively) (Appendix A). All associations were nominally significant.

None of the investigated polymorphisms in antioxidant genes were associated with MRI brain patterns in our study (Table 4, Appendix A).

## 4. Discussion

The present study investigated the association of the common polymorphisms in antioxidant and inflammatory genes with HI-related brain injury in newborns. The most important findings were that polymorphism in *CARD8* was protective in regard to HI brain injury based on the MRI overall findings, and that the polymorphisms in *IL1B* were associated with PLIC, basal ganglia, and white matter MRI abnormalities.

The immune system plays an important role in brain ischemia [40] and inflammasomes are highly expressed in the CNS. An important inflammasome component is CARD8. In our study, *CARD8* rs2043211 was associated with lower overall rates of brain injury and lower brainstem and cortex damage after HIE in newborns treated with HT. To the best of our knowledge, CARD8 and its genetic variability had previously not been studied in terms of outcome following HIE. However, as CARD8 inhibits caspase 1-dependent secretion of IL-1β and activation of nuclear factor NF-κB signaling and thus reduces the inflammatory response [41,42,43], it could contribute to brain injury patterns. Genetic factors could influence the role of CARD8, but the function of the *CARD8* rs2043211 is not fully understood. Up to now it has been known that the SNP rs2043211 results in an A to T transversion that leads to a premature stop codon and produces a truncated CARD8 protein [41,43,44,45] The truncated CARD8 cannot inhibit NF-κB signaling, which results in constitutive production of IL-1β and inflammation [43,45]. This seems to be contrary to our findings, as *CARD8* rs2043211 had a protective role in our study. However, due to alternative splicing, *CARD8* has five known mRNA isoforms that differ in the N-terminal sequence [43].Therefore, the functional consequences of rs2043211 differ among transcripts: in some transcripts, the SNP introduces a stop codon, while in others, it changes the amino acid and affects mRNA stability [43]. The functional role of these isoforms is not yet clear, but it could help explain the contradictory results as the transcripts may vary among the different biological processes [43].

The results of studies on the association between the *CARD8* rs2043211 polymorphism and susceptibility to various diseases are not concordant. Consistent with our results, rs2043211 was associated with a protective role in various diseases where inflammation plays an important role: polycystic ovary syndrome [46], inflammatory bowel disease [47], and Crohn’s disease [45,48]. Moreover, minor rs2043211 T allele had a protective effect in ischemic stroke [41]. Contrary to these findings, in other studies the minor allele was also associated with a higher risk of Alzheimer’s disease in women [49] and with gout [50]. Additionally, studies suggest that decreased activity of CARD8 may delay apoptosis [48,51]. Considering that apoptosis is a key part of the prolonged cell death in HI insult, this could explain the decreased risk of brain injury in hypoxic-ischemic events for *CARD8* rs2043211 T allele. Further studies are therefore needed to determine the role of *CARD8* genetic variability in inflammation. Inflammasomes can be differentially regulated in different tissues. Therefore, it is necessary to evaluate how CARD8 acts in different cell types and brain regions, considering different pathologic processes at the same time [40]. According to the Human Protein Atlas data [52,53], CARD8 has low tissue specificity, also in the brain, but better characterization of different transcripts could help elucidate the specific role of *CARD8* genetic variability in outcome after HIE.

The second important result in our study was that *IL1B* polymorphisms were associated with different brain MRI abnormalities. IL-1β is released as a result of inflammasome activation. In our study, promoter *IL1B* rs1143623 polymorphism was associated with more moderate–severe abnormalities in overall MRI assessment, and newborns with at least one polymorphic allele were more prone to basal ganglia and PLIC injuries based on MRI pattern. Furthermore, promoter *IL1B* rs16944 polymorphism was also associated with more PLIC abnormalities, while *IL1B* rs1071676 located in the 3′ untranslated region was associated with less white matter abnormalities.

IL-1β is a proinflammatory cytokine expressed by various brain cells [54]. It is involved in neuroinflammation and neurodegeneration after HIE [27,55,56]. In previous studies, rs16944 was associated with increased risk for developing cerebral palsy after HIE [27] and increased risk of periventricular leukomalacia after HI insult [57], consistent with our results. *IL1B* rs16944 can affect transcription of the *IL1B* gene as it modifies binding of transcriptional factors [27]. It was associated with increased IL-1β secretion [58] and serum IL-1β levels [59], which could contribute to the observed role of rs16944 after HIE. Additionally, rs16944 was associated with various diseases, including mental disorders and brain activity: hippocampal sclerosis [60], metabolic activity in the dorsolateral prefrontal cortex [61], and grey matter deficits in bipolar disorder [61]. *IL1B* rs1143623 also affects transcription factor binding and serum IL-1β levels [59]. In previous studies it was mostly investigated in various cancers, but recent meta-analysis suggests it does not importantly affect the overall cancer risk [62]. On the other hand, *IL1B* rs1143623 was associated with Alzheimer’s disease pathology [63] and response to biological treatment in psoriasis [64]. The role of *IL1B* rs1071676 is not well established, but it could affect miRNA binding [65], and it was associated with response to anti-TNF therapy [66]. Previous findings are therefore consistent with our results as we have also shown an association between *IL1B* rs1143623 and *IL1B* rs16944 and more basal ganglia and PLIC abnormalities based on MRI pattern.

Central grey matter damage following perinatal HI insult is closely related to the severity of the motor impairment, and studies showed that injury to the basal ganglia-thalamus is a predictor of cerebral palsy, and abnormal PLIC signal intensity on early MRI was associated with the inability to walk independently by two years [67]. So far it is not known how and to which extent polymorphisms have an influence in different regions of the brain, therefore we cannot explain why some *IL1B* polymorphisms have a region-specific effect. However, some studies have already investigated the effect of *IL1B* polymorphisms on specific brain regions by brain MRI. *IL1B* rs16944 allele carriers had smaller hippocampal regions, lower prefrontal functional connectivity, and larger white matter hyperintensity [68]. According to our study, a plausible explanation is that the lesion pattern, which resulted from the pathophysiological mechanism of the HI event, may be responsible for the regional activation of the inflammatory pathway including IL-1β.

In this study, polymorphisms in the common antioxidant genes involved in the processes after HI did not have a major role in brain injury patterns. Only *CAT* rs1001179 T allele had a protective effect on MRI brain injury. This polymorphism alters the transcription factor binding site in the promoter region [17], however, its functional effect is not clear [18]. In our previous study in children with HIE who did not undergo hypothermia, it was associated with the development of cerebral palsy [18]. We could thus hypothesize the role of CAT could be affected by the hypothermia treatment. Some studies have also shown a protective effect of the minor rs1001179 T allele: it was associated with the later presentation of neurological manifestations of Wilson’s disease [69] and with decreased acoustic neuroma risk [70]. None of the other common polymorphisms in the antioxidant pathway were associated with MRI overall assessment or MRI pattern in our study population.

Our study demonstrated the association of some polymorphisms in the antioxidant and inflammatory pathways and MRI abnormalities in newborns after HI insult. In a newborn, HI insult during the phase of primary energy failure and the impairment of cerebral blood flow entails inadequate supply of oxygen and metabolic substrate to the brain tissue, which initiates a cascade of cellular injury [3]. The secondary energy failure phase after HI insult when reperfusion occurs is multifactorial and is characterized by inflammation, cytotoxic oedema, mitochondrial failure, and generation of ROS and free radicals and programmed cell death [3]. The injured brain stimulates innate immune response through activation of microglia and circulating leukocytes which release various molecules including ROS, proteases, and proinflammatory cytokines. The severity of these reactions has been related to cell death and severe neurologic disabilities [71]. HT has been shown to reduce apoptosis, which is an energy-demanding process: higher temperature means higher metabolic rate and faster cellular death. Furthermore, lowering the temperature decreases the metabolic rate of all cell processes and so spares some mitochondrial energy production for reparative mechanisms and survival of cells [5]. During 72 h of HT, more neurons might have a chance to repair the damage of inflammatory and oxidative processes. This would explain reduced disability in children treated with HT after HIE, especially whole-body HT. However, some children among them still have poor outcome [6,20]. The latent period in which HT is most effective is also related to the severity of HIE, as severe HIE may shorten the period and consequently the length of the therapeutic window [72]. Interindividual variability in response to HT may therefore depend on the duration of the ischemic event, the extent of the injury, or genetic factors that alter the subsequent processes of tissue inflammation and antioxidant response following HIE. In our study we found that in addition to HI event, genetic factors may affect the development and patterns of brain injury. In this regard, our results emphasize that genetic modification of the inflammatory response after HI insult is more important for severity of brain injury compared to polymorphisms in the antioxidant pathway.

The strength of our study is that, to the best of our knowledge, it is the first study using a pathway-based approach to address the relationship between some of the inflammation and antioxidant polymorphisms in newborns with HI insult. We were able to demonstrate the association between the common antioxidant and inflammatory gene polymorphisms and MRI brain patterns and the severity of brain damage after HIE. Nevertheless, the results should be interpreted with caution due to retrospective study design and small study sample, which are the main limitations of our study. The retrospective study design prevented us from including all newborns in our medical center treated with HT in the newborn period as some newborns died, several parents did not agree to take part in the study, and for some, contact data were not available. So, with the limited number, the statistical power to detect the association may have been impaired. Therefore, negative results were difficult to address, as we could not detect the polymorphisms’ potential smaller effect sizes. Additional, more comprehensive studies are therefore needed to validate these results. Furthermore, our study did not include newborns from the era before hypothermia. Moreover, in future studies it could be of interest to analyze association between polymorphisms in the antioxidant and inflammatory pathways and the long-term outcome after HI insult in the newborn period since persistent inflammation and epigenetic changes which are crucial in brain repairing and regenerating have been proposed as mechanisms of long-lasting injurious processes.

## 5. Conclusions

The results of this study present a possible association of the polymorphisms in inflammation and the antioxidant pathway with the risk severity of brain injury after HI insult in the newborn period. *CARD8* rs2043211, *IL1B* rs1143623, *IL1B* rs16944, *IL1B* rs1071676, and *CAT* rs1001179 were associated with MRI brain abnormalities. The results of this study are the first indication that individual genetic differences in inflammation and oxidative stress pathways may be involved in the pathogenesis of HI insult in the newborn period.

## Figures and Tables

**Table 1 antioxidants-10-00096-t001:** Clinical characteristics of 44 newborns with HIE.

Clinical Characteristic	Category	N (%)
Gender	Male	25 (56.8)
	Female	19 (43.2)
Gestational age [weeks]	Median (25–75%)	40 (38–40)
Birthweight [g]	Median (25–75%)	3300 (2800–3492.5)
Occipital circumference [cm]	Median (25–75%)	34 (33–35.5)
Mode of delivery	Vaginal	15 (34.9) [1]
	Caesarean section	25 (58.1)
	Vacuum extraction	3 (7.0)
Apgar score in 5 min	≤5	35 (83.3)
	>5	7 (16.7)
Apgar score in 10 min	≤5	18 (46.2) [5]
	>5	21 (53.8)
Sarnat and Sarnat score	2	21 (47.7)
	3	23 (52.3)
Neonatal convulsions	No, N (%)	19 (45.2) [2]
	Yes, N (%)	23 (54.8)

Number of subjects with missing data is presented in [] brackets.

**Table 2 antioxidants-10-00096-t002:** MRI brain patterns and the levels of injury according to Rutherford classification [35].

MRI Brain Pattern	Category	N (%)
Overall assessment	Normal	9 (20.5)
	Mild	17 (38.6)
	Moderate	9 (20.5)
	Severe	9 (20.5)
Basal ganglia	Normal	19 (43.2)
	Mild	9 (20.5)
	Moderate	11 (25.0)
	Severe	5 (11.4)
PLIC	Normal	24 (54.6)
	Equivocal	13 (29.5)
	Loss	7 (15.9)
Brainstem	Normal	33 (75.0)
	Mild	2 (4.5)
	Moderate	5 (11.4)
	Severe	4 (9.1)
White matter	Normal	13 (29.5)
	Mild	13 (29.5)
	Moderate	12 (27.3)
	Severe	6 (13.6)
Cortex	Normal	31 (70.5)
	Mild	3 (6.8)
	Moderate	3 (6.8)
	Severe	7 (15.9)

Legend: MRI = magnetic resonance imaging, PLIC = posterior limb of internal capsule.

**Table 3 antioxidants-10-00096-t003:** Association of common polymorphisms with overall MRI assessment.

Gene	SNP	Role	Genotype	Normal	Mild	Moderate	Severe	*p* *	OR (95% CI) **	*p* **
***Antioxidant pathway***										
***SOD2***	**rs4880**	p.Ala16Val	CC	2 (15.4)	5 (38.5)	2 (15.4)	4 (30.8)		Ref.	
			CT + TT	7 (22.6)	12 (38.7)	7 (22.6)	5 (16.1)	0.804	0.74 (0.20–2.73)	0.647
***CAT***	**rs1001179**	c.-262C > T	CC	2 (6.9)	13 (44.8)	7 (24.1)	7 (24.1)		Ref.	
			CT + TT	7 (46.7)	4 (26.7)	2 (13.3)	2 (13.3)	**0.034**	0.39 (0.10–1.51)	0.173
***GPX1***	**rs1050450**	p.Pro198Leu	CC	6 (31.6)	6 (31.6)	3 (15.8)	4 (21.1)		Ref.	
			CT + TT	3 (12)	11 (44)	6 (24)	5 (20)	0.441	1.35 (0.40–4.57)	0.633
***Inflammatory pathway***										
***NLRP3***	**rs35829419**	p.Gln705Lys	CC	6 (15.8)	15 (39.5)	8 (21.1)	9 (23.7)		Ref.	
			CA	3 (50)	2 (33.3)	1 (16.7)	0 (0)	0.284	0.25 (0.03–2.32)	0.221
***CARD8***	**rs2043211**	p.Cys10Ter	AA	3 (14.3)	6 (28.6)	4 (19)	8 (38.1)		Ref.	
			AT + TT	6 (26.1)	11 (47.8)	5 (21.7)	1 (4.3)	**0.048**	0.26 (0.07–0.94)	**0.040**
***IL1B***	**rs1143623**	c.-1560G > C	GG	7 (30.4)	10 (43.5)	2 (8.7)	4 (17.4)		Ref.	
			GC + CC	2 (9.5)	7 (33.3)	7 (33.3)	5 (23.8)	0.115	3.78 (1.06–13.45)	**0.040**
	**rs16944**	c.-598T > C	CC	7 (30.4)	9 (39.1)	3 (13)	4 (17.4)		Ref.	
			TC + TT	2 (9.5)	8 (38.1)	6 (28.6)	5 (23.8)	0.328	2.51 (0.73–8.63)	0.143
	**rs1071676**	c.*505G > C	GG	1 (5.3)	8 (42.1)	5 (26.3)	5 (26.3)		Ref.	
			GC + CC	8 (32)	9 (36)	4 (16)	4 (16)	0.201	0.42 (0.12–1.45)	0.172
***TNF***	**rs1800629**	c.-308 G > A	GG	7 (22.6)	11 (35.5)	8 (25.8)	5 (16.1)		Ref.	
			GA + AA	2 (15.4)	6 (46.2)	1 (7.7)	4 (30.8)	0.414	0.87 (0.23–3.26)	0.831

* Fisher exact test, comparison of all 4 groups. ** Logistic regression, comparison between moderate + severe vs. normal + mild. Legend: CI = confidence interval, OR = odds ratio, PLIC = posterior limb of internal capsule, SNP = single-nucleotide polymorphism.

**Table 4 antioxidants-10-00096-t004:** The association of polymorphisms with MRI brain patterns.

			Basal Ganglia				PLIC				White Matter			
Gene	SNP	Genotype	Normal + Mild N (%)	Moderate + Severe N (%)	OR (95% CI)	*p*	Normal N (%)	Equivocal or Loss N (%)	OR (95% CI)	*p*	Normal + Mild N (%)	Moderate + Severe N (%)	OR (95% CI)	*p*
***SOD2***	**rs4880**	CC	6 (46.2)	7 (53.8)	Ref.		6 (46.2)	7 (53.8)	Ref.		8 (61.5)	5 (38.5)	Ref.	
		CT + TT	22 (71)	9 (29)	0.35 (0.09–1.34)	0.125	18 (58.1)	13 (41.9)	0.62 (0.17–2.28)	0.471	18 (58.1)	13 (41.9)	1.16 (0.31–4.35)	0.831
***CAT***	**rs1001179**	CC	17 (58.6)	12 (41.4)	Ref.		14 (48.3)	15 (51.7)	Ref.		16 (55.2)	13 (44.8)	Ref.	
		CT + TT	11 (73.3)	4 (26.7)	0.52 (0.13–2.01)	0.340	10 (66.7)	5 (33.3)	0.47 (0.13–1.71)	0.250	10 (66.7)	5 (33.3)	0.62 (0.17–2.26)	0.464
***GPX1***	**rs1050450**	CC	13 (68.4)	6 (31.6)	Ref.		13 (68.4)	6 (31.6)	Ref.		11 (57.9)	8 (42.1)	Ref.	
		CT + TT	15 (60)	10 (40)	1.44 (0.41–5.07)	0.566	11 (44)	14 (56)	2.76 (0.79–9.61)	0.111	15 (60)	10 (40)	0.92 (0.27–3.08)	0.888
***NLRP3***	**rs35829419**	CC	23 (60.5)	15 (39.5)	Ref.		20 (52.6)	18 (47.4)	Ref.		21 (55.3)	17 (44.7)	Ref.	
		CA	5 (83.3)	1 (16.7)	0.31 (0.03–2.89)	0.302	4 (66.7)	2 (33.3)	0.56 (0.09–3.40)	0.525	5 (83.3)	1 (16.7)	0.25 (0.03–2.32)	0.221
***CARD8***	**rs2043211**	AA	11 (52.4)	10 (47.6)	Ref.		10 (47.6)	11 (52.4)	Ref.		11 (52.4)	10 (47.6)	Ref.	
		AT + TT	17 (73.9)	6 (26.1)	0.39 (0.11–1.38)	0.143	14 (60.9)	9 (39.1)	0.58 (0.18–1.94)	0.379	15 (65.2)	8 (34.8)	0.59 (0.17–1.97)	0.389
***IL1B***	**rs1143623**	GG	18 (78.3)	5 (21.7)	Ref.		17 (73.9)	6 (26.1)	Ref.		14 (60.9)	9 (39.1)	Ref.	
		GC + CC	10 (47.6)	11 (52.4)	3.96 (1.07–14.67)	**0.039**	7 (33.3)	14 (66.7)	5.67 (1.54–20.79)	**0.009**	12 (57.1)	9 (42.9)	1.17 (0.35–3.89)	0.802
	**rs16944**	CC	17 (73.9)	6 (26.1)	Ref.		16 (69.6)	7 (30.4)	Ref.		13 (56.5)	10 (43.5)	Ref.	
		TC + TT	11 (52.4)	10 (47.6)	2.58 (0.73–9.12)	0.143	8 (38.1)	13 (61.9)	3.71 (1.06–12.97)	**0.040**	13 (61.9)	8 (38.1)	0.80 (0.24–2.67)	0.717
	**rs1071676**	GG	10 (52.6)	9 (47.4)	Ref.		9 (47.4)	10 (52.6)	Ref.		7 (36.8)	12 (63.2)	Ref.	
		GC + CC	18 (72)	7 (28)	0.43 (0.12–1.51)	0.190	15 (60)	10 (40)	0.60 (0.18–2.00)	0.406	19 (76)	6 (24)	0.18 (0.05–0.68)	**0.011**
***TNF***	**rs1800629**	GG	20 (64.5)	11 (35.5)	Ref.		17 (54.8)	14 (45.2)	Ref.		18 (58.1)	13 (41.9)	Ref.	
		GA + AA	8 (61.5)	5 (38.5)	1.14 (0.30–4.33)	0.851	7 (53.8)	6 (46.2)	1.04 (0.28–3.82)	0.952	8 (61.5)	5 (38.5)	0.87 (0.23–3.26)	0.831

Legend: CI = confidence interval, OR = odds ratio, PLIC = posterior limb of internal capsule, SNP = single-nucleotide polymorphism.

## Data Availability

All the data is presented within the article and in Appendix A. Any additional information is available on request from the corresponding author.

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
