# Peer review of "CARD8* and *IL1B* Polymorphisms Influence MRI Brain Patterns in Newborns with Hypoxic-Ischemic Encephalopathy Treated with Hypothermia"

_antioxidants, 2021, doi:10.3390/antiox10010096_

Round 1

Reviewer 1 Report

The authors have examined the association between the incidence of polymorphisms in select genes related to cell antioxidant and inflammatory pathways in newborns with hypoxic-ischemic encephalopathy. This is a potentially important question but a very difficult one to address adequately and the authors are to be congratulated for their effort. Their results indicate that polymorphisms in inflammatory but not hypoxia pathways is associated with severity of brain injury.  However, there are significant weaknesses that greatly diminish the impact of the report.

Specific comments:

1.  The authors selected 3 antioxidant genes and several inflammatory genes. While the logic in selecting these is good, it is not possible to extrapolate beyond the ones selected for oxidant stress. The system, like a chain, is only as strong as its weakest link. So while these three genes may be normal, if one other key antioxidant gene is defective, antioxidant pathways will be impaired. For example, if glucose-6-phosphate dehydrogenase is defective effectiveness of the glutathione system will be greatly diminished due to inadequate NADPH.  

2.  While the relatively small patient population is adequate if clear positive results are found, it is difficult to reach negative conclusions, such as regarding antioxidant pathways.

3. It is important to state how many patients were excluded based on their exclusion criteria.

Author Response

Subject: Antioxidants. Title: CARD8 and IL1B polymorphisms influence MRI brain patterns in newborns with hypoxic-ischemic encephalopathy treated with hypothermia

Dear Editor,

We would like to thank you for considering our manuscript "CARD8 and IL1B polymorphisms influence MRI brain patterns in newborns with hypoxic-ischemic encephalopathy treated with hypothermia” for publication in your journal.

We addressed the questions raised by the reviewers. We appreciate your comments and questions, which have helped us to improve our paper.

We hope that you will find the new version of the manuscript to be within your expectations.

Yours sincerely,

Vita Dolžan, Aneta Soltirovska Šalamon

Questions raised by the Reviewers

 Reviewer 1

Thank you very much for your comments. We provide the following answers below.

 Q1: The authors selected 3 antioxidant genes and several inflammatory genes. While the logic in selecting these is good, it is not possible to extrapolate beyond the ones selected for oxidant stress. The system, like a chain, is only as strong as its weakest link. So while these three genes may be normal, if one other key antioxidant gene is defective, antioxidant pathways will be impaired. For example, if glucose-6-phosphate dehydrogenase is defective effectiveness of the glutathione system will be greatly diminished due to inadequate NADPH.

A1: Due to the small number of patients our study had to focus on common functional polymorphisms. We have used a pathway based approach to choose genes that are directly involved in the antioxidant defence and in signalling processes of inflammation. Within these genes we have chosen common functional single nucleotide polymorphisms (SNPs) with experimentally or in-silico predicted and established functions to be able to mechanistically explain the potential associations. We agree that glucose-6-phosphate dehydrogenase (G6PD) deficiency greatly diminishes the effectiveness of the glutathione dependent antioxidant system, however in Central and Northern European countries, where malaria was historically not endemic, the incidence of G6PD deficiency is less than 0.1% (Matsouka et al, J Hum Genet. 2005). This is a reason why G6PD or other less common genetic variants were not included in our study.

 Q2: While the relatively small patient population is adequate if clear positive results are found, it is difficult to reach negative conclusions, such as regarding antioxidant pathways.

A2: Indeed, in our study population, statistical power may have been impaired due to a relative small number of patients. Consequently, the impact of the genetic polymorphisms with smaller effect sizes was difficult to detect. Taking your comment into consideration, we have additionally included this limitation as a part of the discussion (rows 316-318): “So, with the limited number, the statistical power to detect the association may have been impaired. Therefore, negative results were difficult to address, as we could not detect the polymorphisms’ that have potential smaller effect sizes.”

According to this, we have emphasized in the discussion that even though that in our study population the selected polymorphisms in the antioxidant genes did not have a major role in brain injury patterns, further studies including a larger number of patients are needed to confirm these results.

Q3. It is important to state how many patients were excluded based on their exclusion criteria.

A3: We have added the number of the excluded patients in the methods section of the manuscript (rows 96-99): “Among 84 children who were treated with total body hypothermia at our centre, 40 were excluded due to the following reasons: newborns did not undergo MRI imaging, informed consent could not be obtained due to death, rejection of parents or legal guardians to participate in the study or insufficient contact data availability”.

Reviewer 2 Report

The objective of this publication is to verify the impact of selected single nucleotide polymorphisms n the genes involved in inflammation and response to oxidative stress on the newborn brain injury after perinatal hypoxic-ischemic insult based on the severity and pattern of magnetic resonance imaging findings. The study is carried out on buccal swabs obtained from 44 patients. the results obtained show, in general, a possible association between genetic variability in inflammation- and antioxidant-related pathways and the severity of brain injury after HI insult in the newborns.

I consider it an interesting article and that it provides information that may have clinical value.

I believe that the antecedents are correct, although I think a little more information could be given about the treatment of hypothermia and on the possible mechanisms that try to explain why it is not totally effective in some cases. In addition, there are two treatments, the selective and the total, in this study the total has been used, but it would be convenient to indicate the existence of both treatments.

Why have these polymorphisms been selected and not others?

Although it is a retrospective study, which is a major limitation, was it not possible to obtain any biochemical data related to greater or lesser inflammation in these children? It would have been interesting to know if there was a greater or less inflammation or a greater or less oxidative aggression. This would give strength to the conclusions obtained.

I really believe that there are multiple limitations in the study, as indicated by the authors, but also that the information provided is interesting and could have clinical application, although it needs more research.

Author Response

 Subject: Antioxidants. Title: CARD8 and IL1B polymorphisms influence MRI brain patterns in newborns with hypoxic-ischemic encephalopathy treated with hypothermia

Dear Editor,

We would like to thank you for considering our manuscript "CARD8 and IL1B polymorphisms influence MRI brain patterns in newborns with hypoxic-ischemic encephalopathy treated with hypothermia” for publication in your journal.

We addressed the questions raised by the reviewers. We appreciate your comments and questions, which have helped us to improve our paper.

We hope that you will find the new version of the manuscript to be within your expectations.

Yours sincerely,

Vita Dolžan, Aneta Soltirovska Šalamon

Reviewer 2

Thank you very much for your comments. We have considered your questions as follows.

Q1: I believe that the antecedents are correct, although I think a little more information could be given about the treatment of hypothermia and on the possible mechanisms that try to explain why it is not totally effective in some cases. In addition, there are two treatments, the selective and the total, in this study the total has been used, but it would be convenient to indicate the existence of both treatments.

A1: With regard to the possible mechanisms that render the hypothermia treatment less effective in some of the newborns we would like to point out that the damage to the newborn's brain after HI event may depend on the duration of the ischemic event itself, the extent of the brain injury, or/and on the genetic variability. Some types of the brain injury could thus be more prone to the treatment due to the difference in the pathophysiological processes at the tissue and cell level. Furthermore, genetic factors could, for example, alter the subsequent processes of tissue inflammation and antioxidant response following HIE. Consequently, this alteration may contribute to the clinical outcome after HIE and lead to different patterns and severity of brain injury seen on the MRI. As it is believed that hypothermia treatment exerts its effect by modifying the processes in which also our selected polymorphisms are implicated, we could presume this could be one of the possible mechanisms of its non- effectiveness (added to Introduction, lines 64 and 65 and to Discussion, lines 298-304).

We have also included the information about the two methods of cooling into the Introduction (lines 40-43): “Two methods of cooling have been used in clinical trials—selective head cooling and whole-body cooling. Whole-body cooling is recommended preferentially because it has less side effects, is easier to set up and use, is less expensive, provides better access to EEGs and is more readily available. “

Q2: Why have these polymorphisms been selected and not others?

A2: We have used a pathway-based approach to select previously studied and functionally characterized genes that are directly involved in the antioxidant defense and in signaling processes of inflammation. Within these genes we have chosen common SNPs with experimentally or in-silico predicted and established functions. We agree that there are other genes and SNPs worth analysing, however, due to the small number of the patients, a larger number of genes or polymorphisms would also largely increase the risk of false positive results.

 Q3: Although it is a retrospective study, which is a major limitation, was it not possible to obtain any biochemical data related to greater or lesser inflammation in these children? It would have been interesting to know if there was a greater or less inflammation or a greater or less oxidative aggression. This would give strength to the conclusions obtained.

 A3: We agree that comparison of biochemical markers of inflammation and oxidative stress would be of great importance. Due to the retrospective nature of the study, buccal swab samples were obtained from the patients for extraction of genomic DNA for genotyping analysis. No biological specimens were available for these patients from the neonatal period that would allow us to obtain relevant biochemical data indicating the oxidative stress or inflammation levels. This is also why we have used MR imaging and scoring to assess the pattern and severity of brain injury within the first week after birth.

In our cohort of children, some biochemical markers of tissue hypoxia and inflammation in newborns with HIE have been analyzed routinely as a part of the therapeutic hypothermia protocol since 2010. As we also included children who underwent treatment with hypothermia before 2010, when biochemical markers of inflammation were also part of clinical management but not assessed in all children at the exact same time points, we decided not to include them in the statistical analysis.

Based on the data from the literature, various biomarkers, e.g. neuron-specific enolase, S100B, lactate, lactate dehydrogenase, glial fibrillary acidic protein, interleukin-8, and vascular endothelial growth factor (Beken S, Turk J Pediatr 2014, Qian J et al. Eur J Pediar 2009, Chalak LF et al. Pediatr 2014, Roka A et al. Acta Paediatr 2012; Massaro AN et al. J Pediatr 2012) were previously associated with the outcome after HIE. However, different results were reported in different studies and reliable early brain-specific serum biomarkers are not available yet.

Round 2

Reviewer 1 Report

The authors have revised the manuscript in response to the previous comments. The softening of conclusions regarding essentially negative results is appropriate.